# Assessment of Ruminating, Eating, and Locomotion Behavior during Heat Stress in Dairy Cattle by Using Advanced Technological Monitoring

**DOI:** 10.3390/ani13182825

**Published:** 2023-09-06

**Authors:** Ramūnas Antanaitis, Karina Džermeikaitė, Agnė Bespalovaitė, Ieva Ribelytė, Arūnas Rutkauskas, Sigitas Japertas, Walter Baumgartner

**Affiliations:** 1Large Animal Clinic, Veterinary Academy, Lithuanian University of Health Sciences, Tilžės Str. 18, LT-47181 Kaunas, Lithuaniaieva.ribelyte@lsmu.lt (I.R.);; 2Practical Training and Research Center, Lithuanian University of Health Sciences, Topolių g. 6, LT-54310 Kaunas, Lithuania; 3University Clinic for Ruminants, University of Veterinary Medicine, Veterinaerplatz 1, A-1210 Vienna, Austria

**Keywords:** heat stress, rumination, eating, activity, innovation, biomarkers

## Abstract

**Simple Summary:**

Heat stress (HS) has a major negative impact on dairy farming. Heat stress in dairy cows can reduce milk production, decrease reproduction rates, and significantly affect animal welfare. Non-invasive methods are commonly used to quantify stress reactions based on changes in behavioral and physiological responses. In this study, a group of nine healthy Lithuanian Black-and-White cows was systematically chosen for the trial, during which their behaviors were recorded using RumiWatch noseband sensors (RWS). We recognized that modern tools like RWS, which integrates a noseband sensor, can be utilized to identify HS and its effects on ruminating, eating, and locomotion behavior during heat stress.

**Abstract:**

Heat stress (HS) significantly impacts dairy farming, prompting interest in precision dairy farming (PDF) for gauging its effects on cow health. This study assessed the influence of the Temperature–Humidity Index (THI) on rumination, eating, and locomotor activity. Various parameters, like rumination time, drinking gulps, chews per minute, and others were analyzed. The hypothesis was that precision dairy farming technology could help detect HS. Nine healthy Lithuanian Black-and-White cows were randomly selected for the trial. RumiWatch noseband sensors recorded behaviors, while SmaXtec climate sensors monitored THI. The data collection spanned from 14 June to 30 June. Cows in the THI class ≥ 72 exhibited reduced drinking time (51.16% decrease, *p* < 0.01), fewer chews per minute (12.9% decrease, *p* < 0.01), and higher activity levels (16.99% increase, *p* < 0.01). THI showed an inverse correlation with drinking time (r = −0.191, *p* < 0.05) and chews per bolus (r = −0.172, *p* < 0.01). Innovative technologies like RumiWatch are effective in detecting HS effects on behaviors. Future studies should explore the impact of HS on RWS biomarkers, considering factors such as lactation stage, number, yield, and pregnancy.

## 1. Introduction

Worldwide warming is described as a long-term and consistent rise in the worldwide yearly temperature. Since the early 1980s, the global temperature has been rising at a rate of about 0.18 °C each decade, implying that, by 2100, the Earth’s temperature might rise by 2.1 °C to 3.9 °C [1]. The rise in global temperature, known as global warming, causes a number of environmental problems. Variation in precipitation as a result of global warming has resulted in increased flooding in some places and increased drought in others [2]. Furthermore, decreased overall snow accumulation has been reported, resulting in a depletion of water resources accessible during the summer months when they are most needed [2]. Hotter and drier conditions are expected to make plants and animals need more water, which will put more pressure on areas that already have trouble getting enough water. This, along with rising temperatures, will cause glaciers to melt and change the way water flows on the surface. High temperatures and extreme events like floods and droughts could lower the quality of water for animals by increasing the amount of pathogens, sediments, salts, nutrients, and phosphorus in the water [3]. Because low soil moisture reduces evaporative cooling from the landscape [4] and high temperatures increase crop water loss [5], high temperatures are frequently associated with water stress. Heat stress (HS) has a significant negative influence on dairy farming. Heat stress in dairy cows can reduce milk production [6], lower reproduction rates [7], impair immunological function [8], and negatively impact animal comfort [9]. Furthermore, climate change is causing more frequent and intense heat extremes, and global surface temperature will continue to rise [10]. HS is presently and will continue to be a significant issue for dairy farmers in the future [11]. Non-invasive approaches are used largely to quantify stress reaction based on changes in behavioral and physiological responses. Technologies for estimating the classical endocrine biomarker cortisol from hair, feces, urine, saliva, and milk could provide useful information for accurately assessing the severity of heat stress in agricultural animals [12]. Smart systems like biosensors and wearable technologies, paired with advanced statistical models like machine learning and technologies like artificial intelligence (AI), can play a significant part in accomplishing the desired goal. The measurement of animal reactions using these techniques ultimately leads to the formation of a set of heat stress thresholds [12]. There are several methods for measuring heat stress. Heat stress in cattle, for example, can be recognized using only environmental temperature since it correlates with rectal temperature [13]. Meteorological variables used to assess heat stress are frequently based on a combination of temperature and relative humidity: the Temperature–Humidity Index (THI), which was first published as a human discomfort index [14]. As a result, the THI can be used to predict HS. THI readings above 72 are stressful for dairy cattle and are likely to reduce their welfare and productivity [15]. The degree of heat stress suffered by cattle is determined by individual traits, environmental conditions, and management tactics [16,17]. HS lowers an animal’s immunity, which has a direct impact on its health and wellbeing [15]. Dairy cows are unable to meet their bodies’ demands for milk production and overall health since HS reduces the amount of energy they consume. Milk production and quality suffer, and the animals become more sensitive to disease [18]. Therefore, appropriate HS reduction strategies in dairy cattle are critical and must be used on farms. Physical adjustments to the cow’s environment (such as providing shade and shelter and cooling cows) and dietary interventions may help to alleviate some of the negative effects of HS and may improve dairy cattle health and output over the summer [15]. According to HS monitoring data, rumination in cows reduces as THI increases [19]. Increased ambient temperature has a direct negative influence on the hypothalamic appetite center, resulting in decreased feed intake. Heat-stressed cows eat less and hence ruminate less, which results in fewer buffering substances entering the rumen; rumination is the primary stimulus for saliva synthesis. Furthermore, as blood flow to the periphery is redistributed (in an attempt to improve heat dissipation) and blood flow to the gastrointestinal tract is reduced, digestive end products (i.e., volatile fatty acids (VFA)) are absorbed less efficiently, resulting in an increased total VFA content in the rumen and thus a decreasing pH. Chronic heat, which causes severe or protracted inappetence, may also result in subclinical rumen acidosis [20]. Furthermore, Weary et al. [21] and Bar and Solomon [22] revealed that animal welfare imbalances, such as heat stress, might be recognized by a decrease in rumination time (RT). However, because RT is positively connected with milk yield [23,24], the desired aim of production is jeopardized by heat stress decreasing RTs in cows [25,26]. We recently discovered that the influence of HS on reticulorumen parameters enhanced the risk of cow acidosis and activity levels. The effects of HS on reticulorumen pH, temperature, and the rumination index were unfavorable. A THI greater than 72 increased the risk of ruminal acidosis and decreased cow physical activity [27]. Precision dairy farming (PDF), also known as the monitoring of behavioral, physiological, or production markers for individual animal sickness, estrus, or comfort, is becoming more popular [28]. PDF is becoming increasingly popular for its uses on livestock farms, both intensive and widespread. PDF has just recently begun to be used, but the necessity of technical support on farms is becoming increasingly vital, allowing its dissemination on farms. A large number of scientific studies on the application of technology, sensors, and computer tools for practically all raised species are available in the literature [29]. PDF must increase the efficiency of its manufacturing systems. To handle data, such systems must undertake information gathering, processing, analysis, and distribution. In terms of grazing lot management, livestock nutrition, and animal health, proper data management can lead to increased productivity [30]. Among studies on the use of monitoring instruments for various animal species are health-monitoring tools, to be used to detect pathologies such as pig coughs, respiratory diseases, and vocalization activities; broad-range tools, to be used for health, welfare, and behavioral factors such as cow rumination rate and heart rate; and early diagnosis of pathologies, to aid intervening before an epidemic breaks out, cutting both costs and the duration of unproductive time to increase the farmer’s capacity [29]. PDF technology can currently track characteristics including laying time, rumination time, movement activity levels, temperature, and milk yield [24]. RumiWatch (RWS) combines a noseband sensor and a pedometer into a single system, resulting in a multipurpose device with exceptional use value, applicability, sensitivity, and specificity [31]. Although the RWS can be used to predict feed intake or evaluate grazing management, it is more expensive than other precision technologies and requires daily maintenance. The RWS provides high-resolution data from two software packages, making it more suitable for study than for practical usage by dairy farmers [31]. According to our knowledge, little research has been conducted to assess the relationship between HS and RWS-registered biomarkers. According to the literature, we hypothesized that the use of precision dairy farming technologies that are registered for ruminating, eating, and locomotion is beneficial in identifying heat stress signals in dairy cattle.

The present study aimed to evaluate the effects that Temperature–Humidity Index has on rumination, eating, and locomotor activity. Rumination and drinking time, drinking gulps, chews per minute, chews per bolus, activity up and down time, and ruminate and eating chews were assessed.

## 2. Materials and Methods

### 2.1. Housing, Animals, and Experimental Design

#### 2.1.1. Housing Conditions and Feeding

The research was carried out between 1 June 2023 and 30 June 2023. Throughout the investigation, all procedures adhered to the Lithuanian Law on Animal Welfare and Protection. After a thorough examination of the methodology by the State Food and Veterinary Service’s Department of Animal Welfare, the trial was approved under the number G2-227. This study was carried out at a public institution practical training and trial center of the Lithuanian University of Health Sciences, on a dairy farm with 120 cows. The cows were housed in an open-stall barn outfitted with rubber mats. Two DeLaval milking robots were used for the milking of 120 cows. The cows were shielded from the sun’s rays, precipitation, wind, and muck, since they were housed in a barn with a full roof and fans that went on automatically when the temperature reached 25 °C. Animals were unable to visit an outside park. A total mixed ration of 50% grain concentrate mash, 18% protein alfalfa hay, 10% grass silage, sugar beet pulp silage, 30% corn silage, 4% grass hay wheat straw, and compound feed was provided to high-producing, multiparous cows. The chemical makeup of the ration was as follows: 47.8% dry matter (DM), 29.02% neutral detergent fiber (DM), 37.8% crude protein (DM), 17.5% acid detergent fiber non-fiber carbohydrates (DM), and 1.8% net lactation energy (DM).

#### 2.1.2. Animals, and Experimental Design

Every day, a local veterinarian checked on the health of these cows. Out of 120, nine clinically healthy cows were randomly assigned to the experiment. Inclusion criteria were Lithuanian Black-and-White breed dairy cows with the following data: an average of 33.5 (±5) days in milk, a second or subsequent lactation, an average of 35 (±5) kg/d, an average body condition score of 3.6 (±0.2) (on a 5-point scale), and a somatic cell count (SCC) level in their milk of less than 195,000/mL (±5.5). Each attribute was retrieved from the farm’s computer system and inputted into a spreadsheet (Delpro DeLaval Inc., Tumba, Sweden). The cows were kept in a loose configuration throughout the year and fed total mixed ration (TMR) at 6 a.m. and 6 p.m. every day, with free access to drinking water.

### 2.2. Measurements

#### 2.2.1. Data of Measurements

The following data were recorded throughout the experiment: rumination, feeding, and movement biomarkers; and Temperature–Humidity Index. Rumination, feeding, and locomotor behavior biomarkers were measured using the RumiWatch noseband sensor (RWS; ITIN + HOCH GmbH, Fütterungstechnik, Liestal, Switzerland) (Figure 1A,B). The RWS is made up of a pressure tube filled with fluids and a noseband halter with an integrated pressure detector. The pressure sensor delivers a pressure signal to the data recorder, which is mounted on the same halter and enclosed in a secure plastic box. A memory card slot and an acceleration sensor for tracking triaxial head movements are also incorporated. The acceleration values and pressure data are recorded as binary files at a frequency of 10 Hz. RumiWatch Manager is connected to the halter by a wireless data transmitter, enabling real-time data collecting. The core algorithms of the RWC software manage the proper classification of behavioral 10 Hz pressure data components in a range of time summaries. The algorithms find unambiguous pressure peak clusters caused by jaw motions and classify them based on their behavioral properties [31].

RWS collected data on ruminating, eating, and locomotor behavior (rumination time, eating time, drinking time, drinking gulps, bolus, chews per minute, chews per bolus, activity up and down time, ruminate chews, and eating chews) (Table 1).

On the farm, a SmaXtec climate sensor recorded the Temperature–Humidity Index (SmaXtec animal care GmbH, Graz, Austria). The humidity index was calculated as THI = 0.8 × T + RH × (T 14.4) + 46.4. A heat stress calculator (SmaXtec animal care GmbH, Graz, Austria) was used to determine the heat index.

#### 2.2.2. Duration of Measurements

RWS was administered on 1 June 2023 and 30 June 2023. The period from 1 June 2023 to 14 June 2023 was an adaptation period, during which cows could adapt to the RWS. The measurement of RWS and THI data began on 14 June 2023 and ended on 30 June 2023. Measurements were always performed every hour, 24 h/day.

### 2.3. Data Analysis and Statistics

The data were analyzed using the IBM SPSS 25.0 software package (SPSS Inc., Chicago, IL, USA). Descriptive statistics and the Kolmogorov–Smirnov test were used to examine normal distributions. The data are shown as the mean and standard deviation of the mean. To identify the statistical links between the analyzed qualities, the Pearson correlation was determined. One-way ANOVA and general linear model—repeated-measures tests (used for repeated measurements, including time periods with the same RumiWatch indication on different experiment days) were used. THI < 72 (comfort zone) and THI ≥ 72 (greater risk for thermal stress) were the two grades. The LSD criterion was employed to compare the mean differences across groups. Significant (*p* < 0.05) was defined as a probability less than 0.05. Descriptive statistics for the examined indicators were based on the THI classifications [32].

## 3. Results

During this study (from 1 June 2023 to 30 June 2023), THI was higher than 72, typically, for two hours per day, between 1:00 p.m. and 3:00 p.m. Our data analysis suggested that cows allocated to the THI class ≥72 had lower average values for DT, CM, and CB, as well as a higher activity level. The DT was 51.16% (*p* < 0.01) and the CM was 12.9% (*p* < 0.01). CB was 50.99% (*p* < 0.01) lower in THI class 72 and Act was 16.99% (*p* < 0.01) higher (Table 2).

THI class was negatively correlated with drinking time (r = −0.191, *p* < 0.05) and with chews per bolus (r = −0.172, *p* < 0.01) (Table 3, Figure 2A,B).

THI was strongly linked with hours (r = 0.432, *p* < 0.01) and weakly with bolus (r = −0.214, *p* < 0.05), chews per minute (r = −0.191, *p* < 0.05), and rumination time (r = −0.212, *p* < 0.05) (Table 4).

Rumination time was found to have a significant, negative relationship with eating time (r = 0.666, *p* < 0.001), eating chews (r = 0.660, *p* < 0.001), drinking gulps (r = 0.671, *p* < 0.001), and activity (r = 0.191, *p* < 0.001). RT was also weakly negatively linked to drinking time (r = 0.172, *p* < 0.05). RT was shown to be substantially related to bolus (r = 0.991, *p* < 0.001) and chews per minute (r = 0.671, *p* < 0.001) (Table 5).

Bolus depended on the daily Temperature–Humidity Index. Analysis of the Temperature–Humidity Index (THI) revealed a strong positive linear connection with hours (r = 0.885, *p* < 0.001). THI had a tendency to increase, on average by 0.885 per hour, *p* < 0.001 (Figure 3A). THI changed from 65.18 to 74.61 during the day, according to a comparison of group means (*p* < 0.001).

Bolus had a weak negative linear connection with hours (r = −0.418, *p* < 0.05). Bolus had a tendency to decrease on average by 0.418 per hour, *p* < 0.05 (Figure 2B). Bolus changed from 0 to 73 during the day, according to a comparison of group means (*p* < 0.05).

## 4. Discussion

There are few data available on the impact of health stress on RWS parameters. The Temperature–Humidity Index (THI) is produced by combining collar measurements with temperature and humidity data, and is crucial for enhancing on-farm decision making. The method serves as the foundation for a completely automated system that allows farmers to assess the efficacy of strategies such as water sprinklers in providing effective relief to their cattle and improving animal wellbeing [33]. Precision livestock farming sensors and daily pattern modelling were effective instruments for monitoring animal behavior and detecting changes caused by heat stress [34]. According to our results, cows with a higher THI (higher than 72) had lower average values for drinking time, chews per minute and chews per bolus. We also found that THI was strongly linked with time (r = 0.432, *p* < 0.01) and weakly with bolus (r = −0.214, *p* < 0.05), chews per minute (r = −0.191, *p* < 0.05), and rumination time (r = −0.212, *p* < 0.05). Mammals employ various mechanisms to maintain thermal homeostasis, often involving valuable resources like glucose or water for survival. Temperature regulation is intricately organized to respond to stimuli such as the environment, reproductive stages, nutrition, and inflammation, with complex neurophysiological pathways intertwined with other systems. Integral to these mechanisms is the skin, which plays a crucial role in detecting thermal changes and microcirculation, enabling heat dissipation or retention through vasodilatation and vasoconstriction. The involvement and integrity of anatomical regions like the cerebral cortex, afferent nerves, and spinal cord are essential for the proper development of thermoregulatory responses, encompassing both physiological and behavioral aspects [35]. Lactating dairy cows require a lot of grain to maintain high milk production [36]. Meeting the energy demands of a high-producing cow is difficult due to appetite loss caused by heat-stress conditions [37]. Furthermore, heat stress can raise metabolic maintenance requirements by 7 to 25%. Heat-stressed cows might enter a negative energy balance due to lower energy availability from their reduced feed intake and increased maintenance expenses [38,39,40]. The negative energy balance that may develop in heat-stressed cows is comparable to, but not identical to, the negative energy balance experienced by cows during early lactation. Early postpartum negative energy balance is associated with an increased risk of metabolic abnormalities and health problems, as well as impaired reproductive performance [41]. THI affects feeding patterns and decreases rumination and resting behaviors, all of which are negative to animal welfare [34]. High THI reduces dry matter intake and rumination duration in dry dairy cows and impacts in situ ruminal degradability, reducing dry matter degradation and the potentially degradable fraction [42]. Animals with a high THI ruminated less and ate more. Given that forage digestion produces a large amount of metabolic heat, raising body temperature, this could be seen as a behavioral adaptation to HS. Cows maintain their thermal balance and mitigate the effects of high external temperatures on heat transfer by limiting their feed intake and avoiding overheating [34]. By lowering rumination, reticulo-rumen motility, and ruminal activity, HS can delay the fractional passage rate of digesta through the gastrointestinal system [43]. HS stimulates the hypothalamus hypothalamic medial satiety area, which inhibits the lateral appetite center, resulting in decreased nutrient and milk consumption. Increased ambient temperature reduces ruminating time and hunger by directly affecting the hypothalamus, which controls appetite [43]. Stress-induced inflammation disrupts feed intake regulation in ruminants compared to homeostatic conditions. Pro-inflammatory cytokines decrease feed intake by affecting hormones that increase or decrease hunger. The direct effects of these cytokines on rumen fermentation and intestinal barriers also affect feed intake indirectly by changing the amount of energy available [44]. Cows eat less and ruminate less during HS, reducing the number of buffering chemicals that reach the rumen (ruminating is the primary activator of saliva production). Rumen acidosis is most likely induced by dietary changes in cows. Cows generally eat 12 to 15 times per day while temperatures are steady; however, when subjected to HS, they only consume 3 to 5 times per day [45]. Chronic heat can result in severe or chronic inappetence, as well as preclinical and acute rumen acidosis [46].

Because forage digestion produces a substantial amount of metabolic heat, which causes an increase in body temperature, this could be considered a behavioral adaptation to heat stress [47,48]. As a result, when external temperatures rise and hamper heat dispersal, cows limit their feed intake to alleviate heat stress [49]. This is similar to the findings looking at daily eating patterns, which show that, under heat stress, animals prefer to shorten their eating time in the afternoon, correlating with higher THI levels. This decrease is partially compensated for at night, when animals under heat stress increase the amount of time they spend eating during the cooler hours of the day. Without heat stress, the distribution of eating time followed a considerably more consistent trend. Three similar peaks were discovered correlating with feeding time after milking, when a significant amount of fresh feed is available [34].

Many studies characterize rumination as a herd indicator that serves as the foundation for the very considerable and developing benefits of breeding as well as reproduction itself. Heat stress and dairy cow health rumination-based interactions are also garnering a lot of attention in recent studies [50]. When ambient climatic conditions were excellent, cattle demonstrated signs of heat stress for extended periods of time during the day (over 6 h) [33]. The number of drinking bouts decreases in the presence of high THI (>82). As a result, it is obvious that heat stress affects drinking habits [50]. Cows gathered around the water trough, panting and refusing to move, and there was a decrease in other voluntary activities such as eating and walking. Brown-Brandl et al. [16] discovered comparable results with feedlot cattle. This could be related to their inability to thermoregulate normally at these THI levels [51].

We found that cows allocated to THI class 72 had a higher activity level. According to the literature, THI increases general activity [34]. Cook et al. previously described how animals increase their activity as THI rises [52]. This lack of rest time is harmful to cows because it reduces blood circulation in their udders [53], reduces milk output [54], and increases the risk of lameness [52]. Reduced resting time increases the risk of lameness in dairy cattle due to the complex physiological consequences of altered resting behavior. Lameness, often caused by conditions like claw horn disruption, is influenced by various factors, including cow-related parameters, housing conditions, and management practices. When cows experience shorter periods of rest, especially in environments with inadequate bedding or space, they may struggle to transition between standing and lying positions, leading to increased stress on the hooves. This stress can result in the development of hoof lesions and inflammatory changes in the third phalanx, making the cows more susceptible to lameness [52].

When THI levels rise, animal welfare suffers as a result of increased activity, changes in feeding patterns, and decreased resting time [34]. A study found that, on hot days, bovine animals exhibited reduced levels of activity in the morning and increased levels of activity in the afternoon [43]. A study conducted by Spanish scientists demonstrated that the resting time of animals suffering from HS was shorter—particularly in the afternoon, where there was a considerable drop in resting time [34]. Heat-stressed cattle stand for longer periods of time to promote heat escape through the skin [45]. Cattle activity level monitoring systems used in modern farms can measure cattle behavior and issue a warning to a computer program if it deviates from normal behavior [46]. Cattle move their heads more under heat stress according to sensors. Cows take more steps per day during the summer months than during the winter months [55]. When changes in feeding, rumination, or standing/lying behaviors are significant enough to justify a more thorough assessment, further wellbeing concerns are discovered [33]. According to the literature, cows relax between 9 and 12 h every day [56]. That is an average of 22 to 30 min every hour. This behavior is a measure of animal welfare in cattle, since it is dramatically affected when the animals are stressed or uncomfortable [57]. According to Provolo and Riva, cows under heat stress spend more time standing in order to obtain more heat dissipation through the skin [45]. This behavior is a useful indicator of cow welfare because it is considerably changed when the animals are anxious or in pain. The amount of time spent resting each hour is reduced when HS occurs [34].

There are limitations of this study, because parameters from RWS are dependent on lactation stage, lactation number, milk yield, and pregnancy [58]. In this present study, we did not investigate the impact of HS on RWS biomarkers considering lactation stage, lactation number, milk yield, or pregnancy. Little research has been done to investigate the relationship between HS and RWS-recorded biomarkers in the literature. Additionally, because in our study we investigated a small number of cows (only nine), we recommend future studies investigating the effect that Temperature–Humidity Index has on rumination, eating, and locomotor activity in larger numbers of animals.

## 5. Conclusions

We studied the influence of HS on ruminating, eating, and locomotion behaviors registered by innovative technologies. HS (THI ≥ 72) had a negative impact on drinking time (51.16% lower, *p* < 0.01) and chews per minute (12.9% lower, *p* < 0.01)) as well as a higher activity level (16.99% higher, *p* < 0.01). THI correlated negatively with RWS-registered drinking time (r = 0.191, *p* < 0.05) and chews per bolus (r = 0.172, *p* < 0.01).

For dairy breeders, implementing PDF technologies like RumiWatch can offer valuable insights into mitigating the impact of HS on cow health. Monitoring THI can help detect HS effects on cow behaviors, such as reduced drinking time, decreased chews per minute, and increased activity levels. To further enhance cow wellbeing, future studies should explore the influence of HS on RumiWatch sensors’ biomarkers, considering factors like lactation stage, number, yield, and pregnancy status with a large number of cows. These findings emphasize the potential of utilizing innovative tools to better manage heat stress and enhance overall herd welfare.

## Figures and Tables

**Figure 1 animals-13-02825-f001:**
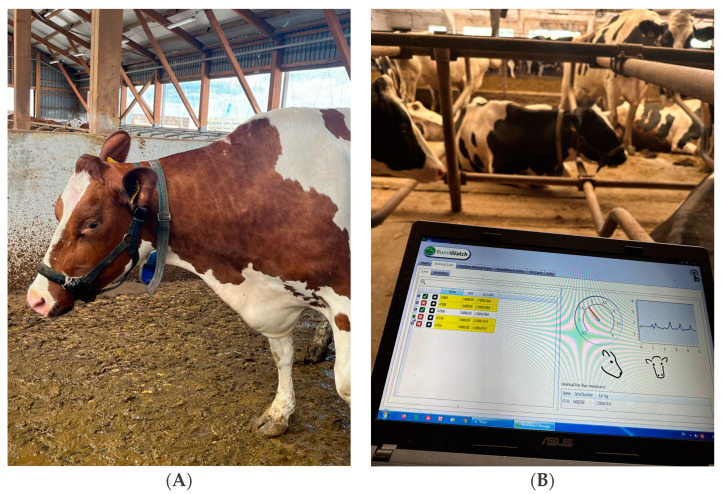
Cow with a RumiWatch noseband sensor (**A**) and data registration (**B**) (RWS; ITIN + HOCH GmbH, Fütterungstechnik, Liestal, Switzerland).

**Figure 2 animals-13-02825-f002:**
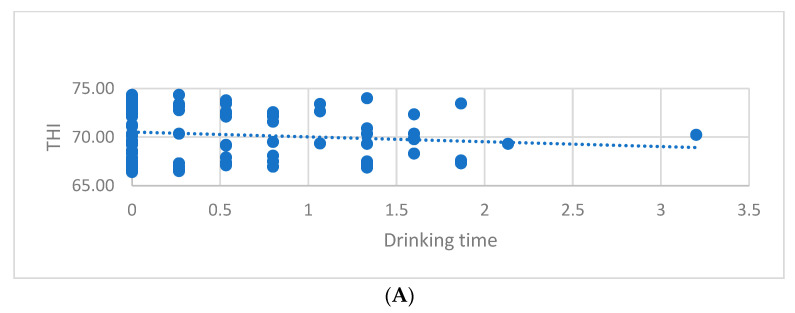
Correlation between THI and drinking time (**A**) and correlation between bolus and hours (**B**). THI—Temperature–Humidity Index; chews per bolus—chews conducted between regurgitation and swallowing of one bolus during rumination.

**Figure 3 animals-13-02825-f003:**
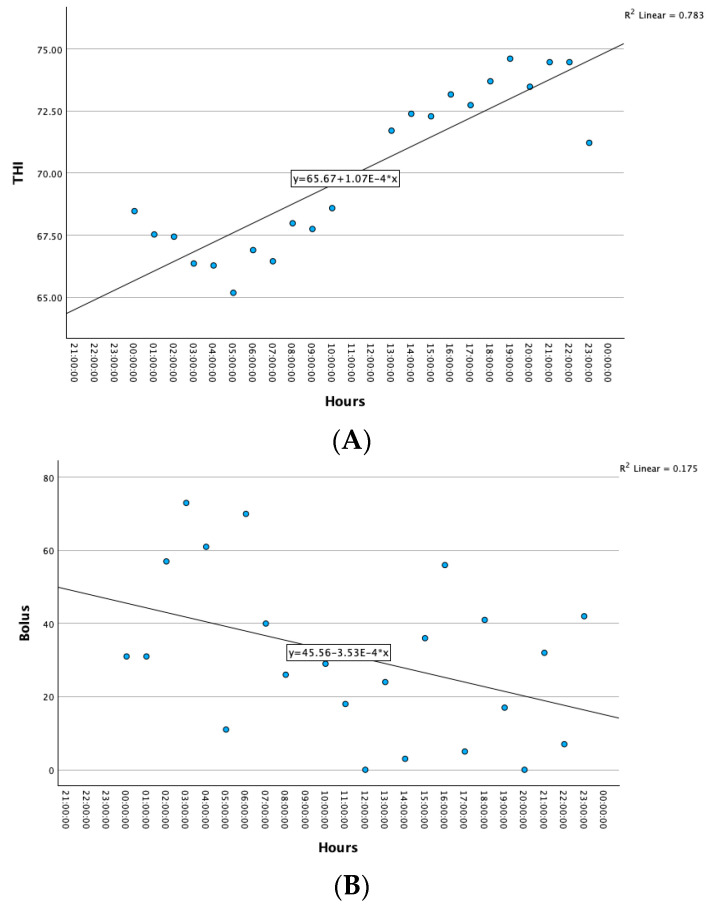
Correlation between THI and hours (**A**) and correlation between bolus and hours (**B**). THI—Temperature–Humidity Index.

**Table 1 animals-13-02825-t001:** Parameters recorded by RWS (ITIN + HOCH GmbH, Fütterungstechnik, Liestal, Switzerland) and their abbreviations and descriptions.

Parameter	Abbreviation	Description
Rumination time	RT	Time spent chewing or ruminating, including 5 s intervals.
Eating time	ET	Time spent chewing or ruminating, including 5 s intervals.
Drinking time	DT	Drinking time, including up to 5 s breaks between gulps
Rumination chews	RC	During rumination, molar chews for mechanical reduction of regurgitated items into smaller masses
Eating chews	EC	Total number of trepidation bites and chews made while eating
Drinking gulps	DG	Total number of gulps consumed while drinking
Bolus	B	Number of boluses consumed during ruminating
Chews per minute	CM	Rumination chews per minute—chews for one minute
Chews per bolus	CB	Chews conducted between regurgitation and swallowing of one bolus during rumination
Activity	Act	The sum of all walking bouts presented as minutes over a specific recording period
Up time	UT	Time spent eating with the head elevated (min/h)
Down time	DT	Feeding time with the head positioned downwards (min/h)

**Table 2 animals-13-02825-t002:** Data analysis according to THI class.

Parameter and Abbreviation	THI Class	N	Mean	Std. Deviation	Std. Error Mean
Rumination time (RT)	THI < 72	86	24.36	14.761	1.592
	THI ≥ 72	62	19.72	15.523	1.971
Eating time (ET)	THI < 72	86	11.99	13.150	1.418
	THI ≥ 72	62	11.54	12.545	1.593
Drinking time (DT)	THI < 72	86	0.43 ^A^	0.687	0.074
	THI ≥ 72	62	0.21 ^B^	0.394	0.050
Rumination chews (RC)	THI < 72	86	1706.94	1103.742	119.020
	THI ≥ 72	62	1571.84	1237.684	157.186
Eating chews (EC)	THI < 72	86	860.13	960.611	103.585
	THI ≥ 72	62	814.81	925.396	117.525
Drinking gulps (DG)	THI < 72	86	643.58	730.619	78.785
	THI ≥ 72	62	624.24	702.293	89.191
Bolus (B)	THI < 72	86	29.36	18.753	2.022
	THI ≥ 72	62	23.76	18.709	2.376
Chews per minute (CM)	THI < 72	86	83.26 ^A^	30.512	3.290
	THI ≥ 72	62	72.51 ^B^	39.595	5.029
Chews per bolus (CB)	THI < 72	86	6.06 ^a^	10.686	1.152
	THI ≥ 72	62	2.97 ^b^	5.008	0.636
Activity (Act)	THI < 72	86	114.60 ^a^	71.832	7.746
	THI ≥ 72	62	138.06 ^b^	84.874	10.779
Up time (UT)	THI < 72	86	13.92	19.569	2.110
	THI ≥ 72	62	11.81	17.705	2.248
Down time (DT)	THI < 72	86	19.85	21.397	2.307
	THI ≥ 72	62	23.73	23.776	3.020

The THI < 72 class has 86 records; the THI ≥ 72 class has 62 records. Means with various superscripts within the same column indicate significant differences between THI groups at the *p* < 0.01 (A, B) or *p* < 0.05 (a, b) levels.

**Table 3 animals-13-02825-t003:** Correlations between THI class, drinking time, and chews per bolus.

Correlations				
		THI Class	Drinking Time (DT)	Chews per Bolus (CB)
THI class	Pearson Correlation	1	−0.191 *	−0.172 *
	Sig. (two-tailed)		0.020	0.036
	N	148	148	148
Drinking time (DT)	Pearson Correlation	−0.191 *	1	0.865 **
	Sig. (two-tailed)	0.020		<0.001
	N	148	148	148
Chews per bolus (CB)	Pearson Correlation	−0.172 *	0.865 **	1
	Sig. (two-tailed)	0.036	<0.001	
	N	148	148	148

* Correlation is significant at the 0.05 level (two-tailed). ** Correlation is significant at the 0.01 level (two-tailed).

**Table 4 animals-13-02825-t004:** Correlations between THI, hours, bolus, chews per minute, and rumination time.

		THI	Hours	Bolus	Chews per Minute	Rumination Time
THI	Pearson Correlation	1	0.432 **	−0.214 **	−0.191 *	−0.212 **
	Sig. (two-tailed)		<0.001	0.009	0.020	0.010
	N	148	148	148	148	148

** Correlation is significant at the 0.01 level (two-tailed). * Correlation is significant at the 0.05 level (2-tailed).

**Table 5 animals-13-02825-t005:** Pearson’s correlations between biomarkers measured by the RumiWatch noseband sensors.

Correlations										
		Rumination Time	Eating Time	Drinking Time	Rumination Chew	Eating Chews	Drinking Gulps	Bolus	Chews per Minute	Activity
Rumination time	Pearson Correlation	1	−0.666 **	−0.172 *	0.734 **	−0.660 **	−0.671 **	0.991 **	0.671 **	−0.655 **
	Sig. (1-tailed)		<0.001	0.018	<0.001	<0.001	<0.001	<0.001	<0.001	<0.001
	N	148	148	148	148	148	148	148	148	148

* Correlation is significant at the 0.05 level (two-tailed). ** Correlation is significant at the 0.01 level (two-tailed).

## Data Availability

The data presented in this study are available within the article.

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
