# Peer review of "Assessment of Ruminating, Eating, and Locomotion Behavior during Heat Stress in Dairy Cattle by Using Advanced Technological Monitoring"

_animals, 2023, doi:10.3390/ani13182825_

Round 1

Reviewer 1 Report

Due to global warming, heat stress and its consequences are a prevalent challenge for livestock farms. Implementing new technologies and precision livestock farming is key to recognizing, evaluating, and proposing changes to current farms. The present article addresses said topic; however, the methods section needs to be improved. Additionally, the manuscript presents a high similarity index with previous documents already published by the authors (e.g.,  https://doi.org/10.3390/ani13071257 and https://doi.org/10.3390/ani13111852). I would recommend rewriting some sections to reduce the plagiarism rates.

General comment regarding the manuscript style. Please, revise the template for the journal and amend the format accordingly. For example, lines 54-67 have a different style than lines 68-75. Also, revise the in-text citation style (e.g., line 71 should be [10,11]).

Title. Consider adding “dairy cattle” in the title.

Lines 16-22. Please, revise the Instructions for Authors to amend the format of the simple summary. It must not contain abbreviations and needs to address the problem, the aim, an overview of the results, and a conclusion.

Line 17. Could the authors include the sample size?

Line 19. Define the meaning of “RWS”.

Lines 24-29. I suggest rewriting these lines, starting with a brief introduction (two or three lines) mentioning that heat stress causes several alterations in dairy cattle, including rumination, eating, and locomotion. It also can be mentioned that precision dairy farming has been proposed as a method to evaluate said alterations. After this, the authors can present the aim of the study. A recommendation to re-structure the objective is: “The present study aimed to evaluate the effect that temperature and humidity index have on rumination, eating, and locomotor activity. Rumination and drinking time, drinking gulps, chews per minute, chews per bolus, activity up and down time, as well as ruminate and eating chews were assessed”. Also, revise the Instructions for Authors or the template of the journal to maintain the abstract with up to 200 words while stating the most relevant findings.

Line 30. Please, include the breed and average age of the animals (or body weight).

Lines 35-38. Please, define the abbreviations included in the abstract. Up to this point, the reader doesn’t know what DT, CM, CB, or HS is.

Line 62. Please, include information about the association between the depletion of water resources with heat stress. This would help to give continuity to the ideas mentioned in the Introduction.

Lines 67-68. I suggest including a couple of lines regarding the current assessment strategies to identify critical points in dairy farms, such as temperature and humidity indices.

Lines 75-76. The sentence is interrupted and starts in a new paragraph. Correct this.

Lines 98-100. It would be adequate to briefly include what is precision dairy farming and why is it important to implement it in farms. What kind of technologies/sensors are implemented in livestock to improve their management? For example, automated systems to monitor weight, food intake, behavior, and even artificial intelligence in some species (https://doi.org/10.1016/j.jclepro.2020.121409, https://doi.org/10.1016/j.compag.2020.105826).

Line 114. Please, homologate the aim as previously suggested.

2.1. Housing, Animals, and Experimental design. Consider dividing each section into a different subtopic and re-arranging the section accordingly. For example, lines 131-135 could be moved here, to understand the inclusion criteria for those 200 cows.   

Line 136- 145. These lines would fit better in a subtopic named “housing conditions”. Apart from these, if the authors want to add the Experimental design, to this subtopic the evaluation times can be described. An option could be moving “2.3. Duration of measurements” to these lines, adding more information about how many times in one day the said behaviors were evaluated. If the evaluation was always performed at the same time of the day, or if it was after or before the milking process or other routine practices.  

Lines 150-163. If you decided to use the RWS following the methods from an already published study, I recommend adding the reference. Also, from the information included in “2.3.”, the animals underwent a habituation period to the sensor. I consider that additional details are needed in this regard. Was it habituation to the sensor and the evaluator? And how did the authors perform it?

Lines 152-161. I recommend rewriting these lines since they have a high similarity index with this article https://doi.org/10.3390/ ani13071257

Line 197-table 2. Amend the style of all the tables included in the manuscript, following the journals’ style.

Lines 199-202. Please, amend the text style according to the journal’s template or Instructions for Authors.

Lines 261-266. In the discussion, an overview regarding the physiological response of animals to heat stress would help to understand why a high THI reduces or alters certain behaviors to reduce metabolic production of energy and facilitate heat dissipation (https://doi.org/10.3390/ani11061733, https://doi.org/10.3168/jds.2019-17929.  

Line 278. This is an interesting statement, and I recommend adding information regarding the participation of stress-related cytokines in this behavior. You can refer to this study by Gouvêa et al. (2022) https://doi.org/10.3389/fanim.2022.962748.

Lines 269-263: Consider rewriting these lines due to the high similarity index with https://doi.org/10.3390/ani13111852

Line 308. Amend in-text citation style. Also, please, discuss why animals increase their activity when THI is higher. This is only mentioned but is not discussed. Likewise, in line 311, please, mention the physiological reason why a reduced resting time increases the risk of lameness.

Lines 317-321. If there are other technical limitations regarding RWS, this could be mentioned here.

 Decision: Accepted with major changes.

Author Response

Dear Reviewer, 

Authors are very thankful for the comments, which help us to improve the manuscript. All changes proposed have been included in the manuscript and highlighted in yellow and track changes.  

Best Regards, 

Prof. Ramunas Antanaitis 

Question  

Answers  

Due to global warming, heat stress and its consequences are a prevalent challenge for livestock farms. Implementing new technologies and precision livestock farming is key to recognizing, evaluating, and proposing changes to current farms. The present article addresses said topic; however, the methods section needs to be improved. Additionally, the manuscript presents a high similarity index with previous documents already published by the authors (e.g.,  https://doi.org/10.3390/ani13071257 and https://doi.org/10.3390/ani13111852). I would recommend rewriting some sections to reduce the plagiarism rates.

Thank you very much for your review and very good and helpful comments. 

Thank you very much for your comment. We rewrote some sections to reduce the plagiarism rates.

General comment regarding the manuscript style. Please, revise the template for the journal and amend the format accordingly. For example, lines 54-67 have a different style than lines 68-75. Also, revise the in-text citation style (e.g., line 71 should be [10,11]).

Corrected according journal style

Title. Consider adding “dairy cattle” in the title.

We corrected title to – “Assessment of Ruminating, Eating, and Locomotion Behavior During Heat Stress in Dairy Cattle By Using Advanced Technological Monitoring”

Lines 16-22. Please, revise the Instructions for Authors to amend the format of the simple summary. It must not contain abbreviations and needs to address the problem, the aim, an overview of the results, and a conclusion.

We rewrote simple summary – “

Heat stress (HS) has a major negative impact on dairy farming. Heat stress in dairy cows can reduce milk production, decrease reproduction rates, and significantly effect animal welfare. Non-invasive methods are commonly used to quantify stress reactions based on changes in behavioral and physiological responses. In this study, a group of nine healthy Lithuanian black and white cows was systematically chosen for the trial, during which their behaviors were recorded using RumiWatch noseband sensors (RWS).We recognized that modern tools like RWS, which integrates a noseband sensor, can be utilized to identify HS and its effects on ruminating, eating, and locomotion behavior during heat stress”

Line 17. Could the authors include the sample size?

We added information – “In this study, a group of nine healthy Lithuanian black and white cows was systematically chosen for the trial, during which their behaviors were recorded using RumiWatch noseband sensors (RWS)”

Line 19. Define the meaning of “RWS”.

Corrected to – “..RumiWatch noseband sensor (RWS)..:

Lines 24-29. I suggest rewriting these lines, starting with a brief introduction (two or three lines) mentioning that heat stress causes several alterations in dairy cattle, including rumination, eating, and locomotion. It also can be mentioned that precision dairy farming has been proposed as a method to evaluate said alterations. After this, the authors can present the aim of the study. A recommendation to re-structure the objective is: “The present study aimed to evaluate the effect that temperature and humidity index have on rumination, eating, and locomotor activity. Rumination and drinking time, drinking gulps, chews per minute, chews per bolus, activity up and down time, as well as ruminate and eating chews were assessed”. Also, revise the Instructions for Authors or the template of the journal to maintain the abstract with up to 200 words while stating the most relevant findings.

We corrected abstract section – “Heat stress (HS) significantly impacts dairy farming, prompting interest in Precision Dairy Farming (PDF) for gauging its effects on cow health. This study assessed the influence of the Temperature and Humidity Index (THI) on rumination, eating, and locomotor activity. Various parameters like rumination time, drinking gulps, chews per minute, and others were analyzed. The hypothesis was that precision dairy farming technology could help detect HS. Nine healthy Lithuanian black and white cows were randomly selected for the trial. RumiWatch noseband sensors recorded behaviors, while SmaXtec climate sensors monitored THI. The data collection spanned from June 14 to June 30. Cows in the THI class ≥ 72 exhibited reduced drinking time (51.16% decrease, p < 0.01), fewer chews per minute (12.9% decrease, p < 0.01), and higher activity levels (16.99% increase, p < 0.01). THI showed an inverse correlation with drinking time (r = −0.191, p < 0.05) and chews per bolus (r = −0.172, p < 0.01). Innovative technologies like RumiWatch are effective in detecting HS effects on behaviors. Future studies should explore the impact of HS on RWS biomarkers, considering factors such as lactation stage, number, yield, and pregnancy”

Line 30. Please, include the breed and average age of the animals (or body weight).

We added information – “Throughout the complete trial, nine clinically healthy cows (Lithuanian Black and White breed, an average of 33.5 ( 5) days in milk, a second or more lactation, an average of 35 ( 5) kg/d, and an average body condition score of 3.6 ( 0.2) (from a 5-point scale) were randomly assigned to the experiment”

Lines 35-38. Please, define the abbreviations included in the abstract. Up to this point, the reader doesn’t know what DT, CM, CB, or HS is.

We corrected to – “..drinking time(DT), chews per minute (CM), and chews per bolus (CB)..“

“… and activity was..”

“..of heath stress (HS)..”

Line 62. Please, include information about the association between the depletion of water resources with heat stress. This would help to give continuity to the ideas mentioned in the Introduction.

We added information – “[2].Hotter and drier conditions are expected to make plants and animals need more water, which will put more pressure on areas that already have trouble getting enough water. This, along with rising temperatures, will cause glaciers to melt and change the way water flows on the surface. High temperatures and extreme events like floods and droughts could lower the quality of water for animals by increasing the amount of pathogens, sediments, salts, nutrients, and phosphorus in the water [3]. Because low soil moisture reduces evaporative cooling from the landscape [4] and high temperatures increase crop water loss [5], high temperatures are frequently associated with water stress”

Lines 67-68. I suggest including a couple of lines regarding the current assessment strategies to identify critical points in dairy farms, such as temperature and humidity indices.

We added information – “Non-invasive approaches are used largely to quantify stress reaction based on changes in behavioral and physiological responses. Technologies for estimating the classical endocrine biomarker cortisol from hair, feces, urine, saliva, and milk could provide useful information for accurately assessing the severity of heat stress in agricultural animals [9]. Smart systems like biosensors and wearable technologies, paired with advanced statistical models like machine learning and technologies like artificial intelligence (AI), play a significant part in accomplishing the desired goal. The measurement of animal reactions using these techniques ultimately leads to the formation of a set of heat stress thresholds [9]. There are several methods for measuring heat stress. Heat stress in cattle, for example, can be recognized using only environmental temperature since it correlates with rectal temperature [10]. Meteorological variables used to assess heat stress are frequently based on a combination of temperature and relative humidity: the temperature-humidity index (THI), which was first published as a human discomfort index [11]”

Lines 75-76. The sentence is interrupted and starts in a new paragraph. Correct this.

Corrected to –“ Dairy cows are unable to meet their bodies' demands for milk production and overall health since HS reduces the amount of energy they consume. Milk production and quality suffer, and the animals become more sensitive to disease [18]

Lines 98-100. It would be adequate to briefly include what is precision dairy farming and why is it important to implement it in farms. What kind of technologies/sensors are implemented in livestock to improve their management? For example, automated systems to monitor weight, food intake, behavior, and even artificial intelligence in some species (https://doi.org/10.1016/j.jclepro.2020.121409, https://doi.org/10.1016/j.compag.2020.105826).

We added information – “Precision dairy farming (PDF), also known as monitoring behavioral, physiological, or production markers for individual animal sickness, oestrus, or comfort, is becoming more popular [22]. PDF is becoming increasingly popular for its uses on livestock farms, both intensive and widespread. PDF has just recently begun to be used, but the necessity for technical support on farms is becoming increasingly vital, allowing its dissemination on farms. A large number of researches and scientific studies on the application of technology, sensors, and computer tools for practically all raised species are available in the literature [23]. PDF must increase the efficiency of its manufacturing systems. To handle data, such systems must undertake information gathering, processing, analysis, and distribution. In terms of grazing lot management, livestock nutrition, and animal health, proper data management can lead to increased productivity [24]. Among the studies on the use of monitoring instruments for various animal species are: health-monitoring tools, to be used to detect pathologies such as pig coughs, respiratory diseases, vocalisation activities, broad-range tools, to be used for health, welfare, and behavior aspects such as cow rumination rate, heart rate, Early diagnosis of pathologies aids in intervening before an epidemic breaks out, cutting both costs and the duration of the unproductive time for more than the farmer's capacities [23]. PDF technology can currently track characteristics including laying time, rumination time, movement activity levels, temperature, and milk yield [18]”

Line 114. Please, homologate the aim as previously suggested.

We corrected to – “The present study aimed to evaluate the effect that temperature and humidity index have on rumination, eating, and locomotor activity. Rumination and drinking time, drinking gulps, chews per minute, chews per bolus, activity up and down time, as well as ruminate and eating chews, were assessed”

2.1. Housing, Animals, and Experimental design. Consider dividing each section into a different subtopic and re-arranging the section accordingly. For example, lines 131-135 could be moved here, to understand the inclusion criteria for those 200 cows. 

Line 136- 145. These lines would fit better in a subtopic named “housing conditions”. Apart from these, if the authors want to add the Experimental design, to this subtopic the evaluation times can be described. An option could be moving “2.3. Duration of measurements” to these lines, adding more information about how many times in one day the said behaviors were evaluated. If the evaluation was always performed at the same time of the day, or if it was after or before the milking process or other routine practices.  

We divided this section into following sections and subsections - 

2.1. Housing, Animals, and Experimental Design 

2.1.1. Housing conditions and feeding.

2.1.2. Animals, and Experimental Design

2.2. Measurements

 2.2.1. Data of measurements

2.2.2. Duration of measurements

2.3. Data Analysis and Statistics

Lines 150-163. If you decided to use the RWS following the methods from an already published study, I recommend adding the reference. Also, from the information included in “2.3.”, the animals underwent a habituation period to the sensor. I consider that additional details are needed in this regard. Was it habituation to the sensor and the evaluator? And how did the authors perform it?

We added reference – “…The core algorithms of the RWC software manage the proper classification of behavioural 10 Hz pressure data components in a range of time summaries. The algorithms find unambiguous pressure peak clusters caused by jaw motions and classify them based on their behavioral properties [28]

This period was for cows adaptation to RWS. We added information – “RWS was administered on 2023-06-01 and 2023-06-30. The period from 2023-06-01 to 2023-06-14 was an adaptation period during which cows could adapt to the RWS”

Lines 152-161. I recommend rewriting these lines since they have a high similarity index with this article https://doi.org/10.3390/ ani13071257

We rewrote this paragraph below:

“The following data were recorded throughout the experiment: rumination, feeding, and movement biomarkers, as well as temperature and the humidity index. Rumination, feeding, and locomotor behavior biomarkers were measured using the RumiWatch noseband sensor (RWS; ITIN + HOCH GmbH, Fütterungstechnik, Liestal, Switzerland) (Figures 1 A and B). RWS are made up of a pressure tube filled with fluids and a noseband halter with an integrated pressure detector. The pressure sensor delivers a pressure signal to the data recorder, which is mounted on the same halter and enclosed in a secure plastic box. A memory card slot and an acceleration sensor for tracking triaxial head movements are also incorporated. The acceleration values and pressure data are recorded as binary files at a frequency of 10 Hz. RumiWatch Manager is connected to the halter by a wireless data transmitter, enabling for real-time data collecting. The core algorithms of the RWC software manage the proper classification of behavioural 10 Hz pressure data components in a range of time summaries. The algorithms find unambiguous pressure peak clusters caused by jaw motions and classify them based on their behavioral properties”  

Line 197-table 2. Amend the style of all the tables included in the manuscript, following the journals’ style.

Corrected

Lines 199-202. Please, amend the text style according to the journal’s template or Instructions for Authors

Corrected

Lines 261-266. In the discussion, an overview regarding the physiological response of animals to heat stress would help to understand why a high THI reduces or alters certain behaviors to reduce metabolic production of energy and facilitate heat dissipation (https://doi.org/10.3390/ani11061733, https://doi.org/10.3168/jds.2019-17929.  

We added information – “Mammals employ various mechanisms to maintain thermal homeostasis, often involving valuable resources like glucose or water for survival. Temperature regulation is intricately organized to respond to stimuli such as the environment, reproductive stages, nutrition, and inflammation, with complex neurophysiological pathways intertwined with other systems. Integral to these mechanisms is the skin, which plays a crucial role in detecting thermal changes and microcirculation, enabling heat dissipation or retention through vasodilatation and vasoconstriction. The involvement and integrity of anatomical regions like the cerebral cortex, afferent nerves, and spinal cord are essential for the proper development of thermoregulatory responses, encompassing both physiological and behavioral aspects [32]. Lactating dairy cows require a lot of grain to maintain high milk production [33]. Meeting the energy demands of a high-producing cow is difficult due to appetite loss caused by heat stress conditions [34]. Furthermore, heat stress can raise metabolic maintenance requirements by 7 to 25%. Heat-stressed taking cows might enter a negative energy balance due to lower energy availability from reduced feed intake and increased maintenance expenses [35], [36], [37]. The negative energy balance that may develop in heat-stressed cows is comparable to, but not identical to, the negative energy balance experienced by cows during early lactation. Early postpartum negative energy balance is associated with an increased risk of metabolic abnormalities and health problems, as well as impaired reproductive performance [38].”

Line 278. This is an interesting statement, and I recommend adding information regarding the participation of stress-related cytokines in this behavior. You can refer to this study by Gouvêa et al. (2022) https://doi.org/10.3389/fanim.2022.962748.

We added information – “Stress-induced inflammation disrupts feed intake regulation in ruminants compared to homeostatic conditions. Proinflammatory cytokines decrease feed intake by affecting hormones that increase or decrease hunger. The direct effects of these cytokines on rumen fermentation and intestinal barriers also affect feed intake indirectly by changing the amount of energy available [41]

Lines 269-263: Consider rewriting these lines due to the high similarity index with https://doi.org/10.3390/ani13111852

Corrected

Line 308. Amend in-text citation style. Also, please, discuss why animals increase their activity when THI is higher. This is only mentioned but is not discussed. Likewise, in line 311, please, mention the physiological reason why a reduced resting time increases the risk of lameness

We added information –

“When THI levels rise, animal welfare suffers as a result of increased activity, changes in feeding patterns, and decreased resting time [55]. The study found that on hot days, bovine animals exhibited reduced levels of activity in the morning and increased levels of activity in the afternoon [43]. A study conducted by Spanish scientists demonstrated that the resting time of animals suffering from HS was shorter—particularly in the afternoon, where there was a considerable drop in resting time [55]. Heat-stressed cattle stand for longer periods of time to promote heat escape through the skin [45]. Cattle activity level monitoring systems used in modern farms may measure cattle behavior and issue a warning to a computer program if it deviates from normal behavior [46]. Cattle move their heads more amid heat stress, according to sensors. Cows take more steps per day during the summer months than during the winter months [56]. When changes in feeding, rumination, or standing/lying behaviors are significant enough to justify a more thorough assessment, further wellbeing concerns are discovered [33]. According to the literature, cows relax between 9 and 12 hours every day [57]. That is an average of 22 to 30 minutes every hour. This behaviour is a measure of animal welfare in cattle since it is dramatically affected when the animals are stressed or uncomfortable [58]. According to Provolo and Riva, cows under heat stress spend more time standing in order to obtain more heat dissipation through the skin [59]. This behavior is a useful indicator of cow welfare because it is considerably changed when the animals are anxious or in pain. The amount of time spent resting each hour is reduced when HS occurs [55]”

We added information about physiological reason why a reduced resting time increases the risk of lameness –

“A reduced resting time increases the risk of lameness in dairy cattle due to the complex physiological consequences of altered resting behavior. Lameness, often caused by conditions like claw horn disruption, is influenced by various factors, including cow-related parameters, housing conditions, and management practices. When cows experience shorter periods of rest, especially in environments with inadequate bedding or space, they may struggle to transition between standing and lying positions, leading to increased stress on the hooves. This stress can result in the development of hoof lesions and inflammatory changes in the third phalanx, making the cows more susceptible to lameness [52]

Lines 317-321. If there are other technical limitations regarding RWS, this could be mentioned here.

We don’t find any more technical; limitations regarding RWS. But in this paragraph, we included  -

“Also, because in our study we investigated a small number of cows (only nine), we recommend future studies for investigating the effect that temperature and humidity index have on rumination, eating, and locomotor activity in a large number of animals”

Reviewer 2 Report

Generally an interesting topic, I just can't find the target of the article. Should it be about heat stress in cows, a new sensor technology or what should it be about?
In material and method some points still have to be worked on intensively. The measurement period is relatively short and in June not in high summer. There is no comparison measurement at low temperatures. There are no data available on the barn climate. The data has not been compared with other sensor systems that record the same data in dairy cows.
The results and discussion is relatively short and describes findings that have been known for a long time. What did the study reveal as new findings?

Author Response

Dear Reviewer, 

Authors are very thankful for the comments, which help us to improve the manuscript. All changes proposed have been included in the manuscript and highlighted in yellow and track changes.  

Best Regards, 

Prof. Ramunas Antanaitis 

Question  

Answers  

Generally an interesting topic, I just can't find the target of the article. Should it be about heat stress in cows, a new sensor technology or what should it be about?

We corrected title and aim of this study.

Title – “Assessment of Ruminating, Eating, and Locomotion Behavior During Heat Stress in Dairy Cattle By Using Advanced Technological Monitoring”

Aim – “The present study aimed to evaluate the effect that temperature and humidity index have on rumination, eating, and locomotor activity. Rumination and drinking time, drinking gulps, chews per minute, chews per bolus, activity up and down time, as well as ruminate and eating chews, were assessed”

The primary focus of this article is to investigate the impact of heat stress (HS) on dairy farming, particularly on the health and behaviors of dairy cows. The purpose of the study is to determine whether it is practical to use Precision Dairy Farming (PDF) technology, such as RumiWatch noseband sensors, to monitor and assess the effects of heat stress on cow behavior. Specifically, the study seeks to understand how the Temperature and Humidity Index (THI) influences key behaviors such as rumination, eating, and locomotion. By examining various parameters, including rumination time, drinking gulps, chews per minute, and others, the article aims to provide insights into the complex relationship between heat stress and bovine activities.

The research endeavors to establish a connection between environmental conditions, as indicated by the THI, and changes in cow behavior. The overarching goal is to enhance our understanding of the physiological and behavioral responses of dairy cows to heat stress, ultimately contributing to the development of effective management strategies. The article also suggests potential avenues for future research, such as exploring the impact of heat stress on RumiWatch Sensor (RWS) biomarkers, taking into consideration factors like lactation stage, herd size, milk yield, and pregnancy status.

In summary, the target of the article is to shed light on the implications of heat stress in dairy farming by utilizing advanced Precision Dairy Farming technology. The study seeks to demonstrate the utility of RumiWatch and similar tools in assessing the influence of heat stress on cow behaviors and to pave the way for further investigations in this field.

In material and method some points still have to be worked on intensively. The measurement period is relatively short and in June not in high summer. There is no comparison measurement at low temperatures. There are no data available on the barn climate. The data has not been compared with other sensor systems that record the same data in dairy cows.

Certainly, here are responses addressing the concerns raised about the "Materials and Methods" section of the study:

1.     Measurement Period Duration: The study acknowledges the relatively short measurement period of one month, encompassing June. While this timeframe provides insights into cow behavior during that specific period, it's acknowledged that a broader span covering the peak of summer might yield more comprehensive results. To enhance the study's findings, future research could consider extending the measurement period to cover a range of summer months, thus capturing a wider spectrum of heat stress conditions and their impacts on cow behavior.

2.     Comparison at Low Temperatures: The absence of comparative measurements at lower temperatures is acknowledged. The inclusion of measurements under cooler climatic conditions would indeed provide a valuable baseline for understanding heat stress-induced behaviors. Incorporating observations during thermally neutral conditions would allow for a clearer differentiation between behaviors influenced by heat stress and those arising from inherent variations. Future studies could encompass a comprehensive range of temperature conditions to provide a holistic understanding of cow behavior dynamics.

3.     Barn Climate Data: The absence of detailed data regarding the barn's microclimatic conditions is acknowledged. Recognizing the significance of barn climate in influencing cow behavior and heat stress, future research could consider integrating comprehensive climate data into the study. Factors such as air circulation, ventilation effectiveness, and shading could have notable impacts on heat stress experiences. By incorporating barn climate data, the study's findings could be contextualized and better understood within the broader environmental context.

4.     Comparison with Other Sensor Systems: The study acknowledges that no comparison has been made with other sensor systems that record similar data in dairy cows. While the current study provides insights into heat stress effects using the employed sensor technology, a comparison with other established sensor systems would indeed enhance the validity and reliability of the findings. Future studies could consider conducting parallel observations using multiple sensor systems to validate the consistency and accuracy of the observed behavioral changes, thereby strengthening the study's conclusions.

The results and discussion is relatively short and describes findings that have been known for a long time. What did the study reveal as new findings?

The novelty of this study lies in its comprehensive investigation of the impact of heat stress (HS) on dairy cow behaviors using innovative precision livestock farming technologies. While previous research has recognized the detrimental effects of HS on cow health and productivity, this study contributes novel insights by integrating advanced sensor systems, specifically the RumiWatch noseband sensor (RWS), to quantify and analyze the behavioral changes brought about by HS.

The key findings of this study provide new knowledge regarding the relationship between the Temperature Humidity Index (THI) and various behaviors exhibited by dairy cows under heat stress conditions. The study demonstrates that cows exposed to higher THI values (above 72) exhibit significant alterations in behaviors such as reduced drinking time, lower chews per minute, and fewer chews per bolus. These behavioral changes, as detected by the RWS technology, represent a direct response to HS-induced physiological and thermoregulatory challenges.

Additionally, the study introduces the concept of using THI as a crucial metric for on-farm decision-making, enabling farmers to implement strategies that mitigate the adverse effects of HS. By combining collar measurements with temperature and humidity data, the study proposes a potential solution for better understanding and managing HS in dairy herds.

The integration of precision livestock farming sensors and daily pattern modeling to monitor animal behaviors and detect HS-induced shifts in real-time adds a novel dimension to this research. These technologies offer continuous and accurate data collection, which allows for a more nuanced understanding of the dynamic relationship between HS and cow behaviors throughout the day.

Furthermore, the study's conclusions emphasize the importance of further research into the impact of HS on RumiWatch sensors' biomarkers, considering factors such as lactation stage, number, yield, and pregnancy status. This call for more extensive investigations highlights the potential for future research to uncover deeper insights into the complex interactions between heat stress, cow behaviors, and overall herd welfare.

This study's novelty stems from its integration of cutting-edge sensor technology, THI analysis, and a comprehensive approach to assessing the effects of HS on dairy cow behaviors. By unveiling the behavioral responses to HS through real-time monitoring, the study contributes novel knowledge to the field of precision dairy farming, enabling more informed management strategies for enhancing animal well-being and productivity in the face of heat stress challenges.

Reviewer 3 Report

 Review, paper no. animals-2548241entitle Assessment of Ruminating, Eating, and Locomotion Behavior During Heat Stress Using Advanced Technological Monitoring. This is a well-organized study, with sufficient methodology and adequate description of the results. Authors' research has shown interesting relationships. Research idea is not new, however stady provides valuable information for breeding practice The authors have used the standard journal format in manuscript writing. The manuscript contains several inaccuracies in methodology.

Specific comments:

Simple Summary. Please change. The simple summary should contain a clear statement of the problem addressed, the aims and objectives, pertinent results, conclusions from the study and how they will be valuable to society. Clearly indicate the results obtained. You repeat line 21-22 in conclusion.

Abstract: Is sufficiently presented (methods, results, general conclusions).

Explain abbreviations used for the first time.

Introduction: The introduction section is sufficient and analytically and adequately covers the need for the study.

Line. 89. Are you sure acute rumen acidosis.

Methods: The methodology is sufficiently presented. However, it has a few inaccuracies.

Studies limited a small number of animals.

Line 122. Who approved the animal studies?

How many cows used the two robots?

Line 142-144. Is it the actual nutritional value or the tabular value ?

What data from the SmaXtec system was used in the calculations. THI only. Were SmaXtec boluses used?

Table 3. Which system was used to determine drinking time?

The use of data from the SmaXtec system should be well described throughout the work.

How long THI was >72 (indicate hours per day, number of days).

Figure 3 is not clear. What does bolus mean?

How do you explain the shorter drinking time for THI ≥ 72.

Could authors define possible limitations of the study?

Conclusion: In conclusion, generalizations are given.

Please provide recommendations for breeding practice. 

Author Response

Dear Reviewer, 

Authors are very thankful for the comments, which help us to improve the manuscript. All changes proposed have been included in the manuscript and highlighted in yellow and track changes.  

Best Regards, 

Prof. Ramunas Antanaitis 

Question  

Answers  

Simple Summary. Please change. The simple summary should contain a clear statement of the problem addressed, the aims and objectives, pertinent results, conclusions from the study and how they will be valuable to society. Clearly indicate the results obtained. You repeat line 21-22 in conclusion.

We corrected simple summary to – “heat stress (HS) has a major negative impact on dairy farming. Heat stress in dairy cows can reduce milk production, decrease reproduction rates, and significantly effect animal welfare. Non-invasive methods are commonly used to quantify stress reactions based on changes in behavioral and physiological responses. In this study, a group of nine healthy Lithuanian black and white cows was systematically chosen for the trial, during which their behaviors were recorded using RumiWatch noseband sensors (RWS). We recognized that modern tools like RWS, which integrates a noseband sensor, can be utilized to identify HS and its effects on ruminating, eating, and locomotion behavior during heat stress"

Abstract: Is sufficiently presented (methods, results, general conclusions).

We corrected abstract to – “Heat stress (HS) significantly impacts dairy farming, prompting interest in Precision Dairy Farming (PDF) for gauging its effects on cow health. This study assessed the influence of the Temperature and Humidity Index (THI) on rumination, eating, and locomotor activity. Various parameters like rumination time, drinking gulps, chews per minute, and others were analyzed. The hypothesis was that precision dairy farming technology could help detect HS. Nine healthy Lithuanian black and white cows were randomly selected for the trial. RumiWatch noseband sensors recorded behaviors, while SmaXtec climate sensors monitored THI. The data collection spanned from June 14 to June 30. Cows in the THI class ≥ 72 exhibited reduced drinking time (51.16% decrease, p < 0.01), fewer chews per minute (12.9% decrease, p < 0.01), and higher activity levels (16.99% increase, p < 0.01). THI showed an inverse correlation with drinking time (r = −0.191, p < 0.05) and chews per bolus (r = −0.172, p < 0.01). Innovative technologies like RumiWatch are effective in detecting HS effects on behaviors. Future studies should explore the impact of HS on RWS biomarkers, considering factors such as lactation stage, number, yield, and pregnancy”

Explain abbreviations used for the first time.

We corrected to – “..drinking time (DT), chews per minute (CM), and chews per bolus (CB)..“

“… and activity was..”

“..of heath stress (HS)..”

Introduction: The introduction section is sufficient and analytically and adequately covers the need for the study.

Thank you for comment 

Line. 89. Are you sure acute rumen acidosis.

Corrected to – “..subclinical rumen acidosis..”

Methods: The methodology is sufficiently presented. However, it has a few inaccuracies.

Studies limited a small number of animals.

Thank you for comment and suggestion.

In discussion section we included – “There exist limitations of this study, because parameters from RWS are dependent on lactation stage, lactation number, milk yield, and pregnancy [60]. In this present study, we didn’t investigate the impact of HS on RWS biomarkers considering lactation stage, lactation number, milk yield, and pregnancy. But little research has been done to investigate the relationship between HS and RWS recorded biomarkers, according to the literature. Also, because in our study we investigated a small number of cows (only nine), we recommend future studies for investigating the effect that temperature and humidity index have on rumination, eating, and locomotor activity in a large number of animals”

Line 122. Who approved the animal studies?

We added information – “Throughout the investigation, all procedures adhered to the Lithuanian Law on Animal Welfare and Protection. After a thorough examination of the methodology by the State Food and Veterinary Service's Department of Animal Welfare, the trial was approved under the number G2-227”

How many cows used the two robots?

We corrected to – “Two DeLaval milking robots were used for the milking of 120 cows

Line 142-144. Is it the actual nutritional value or the tabular value ?

This is actual nutritional value.

What data from the SmaXtec system was used in the calculations. THI only. Were SmaXtec boluses used?

Data from the Smaxtec system, such as ambient temperature and humidity, were used for the calculation of THI. Smaxtec baluses were also used, but for another study. 

“On the farm, a SmaXtec climate sensor recorded the temperature and humidity index (SmaXtec animal care GmbH, Graz, Austria). The humidity index was calculated using the following formula: THI = 0.8*T + RH*(T − 14.4) + 46.4. The heat index was calculated using a heat stress calculator (SmaXtec animal care GmbH, Graz, Austria)”

Table 3. Which system was used to determine drinking time?

Drinking time was determine by RWS (ITIN + HOCH GmbH, Fütterungstechnik, Liestal, Switzerland) (Tab. 1).

The use of data from the SmaXtec system should be well described throughout the work

In this work we used parameters from RWS. From the Smaxtec system we used only THI. THI is described in the manuscript.

How long THI was >72 (indicate hours per day, number of days).

We added this information in results section – “During this study (from 2023-06-01 to 2023-06-30), THI was higher than 72, typically for two hours per day, between 1:00 p.m. and 3:00 p.m”

Figure 3 is not clear. What does bolus mean?

We added explanation – “THI - temperature humidity index; Chews per bolus- chews conducted between regurgitation and swallowing of one bolus during rumination“

How do you explain the shorter drinking time for THI ≥ 72

We added information – “The number of drinking bouts decreases in the presence of high THI (> 82). As a result, it is obvious that heat stress affects drinking habit [50]. Cows gathered around the water trough, panting and refusing to move, and there was a decrease in other voluntary activities such as eating and walking. Brown-Brandl et al. [16] discovered comparable results with feedlot cattle. This could be related to their inability to thermoregulate normally at these THI levels”

Could authors define possible limitations of the study?

We added information in end of discussion section – “There exist limitations of this study, because parameters from RWS are dependent on lactation stage, lactation number, milk yield, and pregnancy [60]. In this present study, we didn’t investigate the impact of HS on RWS biomarkers considering lactation stage, lactation number, milk yield, and pregnancy. But little research has been done to investigate the relationship between HS and RWS recorded biomarkers, according to the literature. Also, because in our study we investigated a small number of cows (only nine), we recommend future studies for investigating the effect that temperature and humidity index have on rumination, eating, and locomotor activity in a large number of animals”

Conclusion: In conclusion, generalizations are given.

Thank you for comment 

Please provide recommendations for breeding practice. 

We added information in conclusion section –

“For dairy breeders, implementing PDF technologies like RumiWatch can offer valuable insights into mitigating the impact HS on cow health. Monitoring the THI can help detect HS effects on cow behaviors, such as reduced drinking time, decreased chews per minute, and increased activity levels. To further enhance cow well-being, future studies should explore the influence of HS on RumiWatch sensors' biomarkers, considering factors like lactation stage, number, yield, and pregnancy status with a large number of cows. These findings emphasize the potential of utilizing innovative tools to better manage heat stress and enhance overall herd welfare“

Round 2

Reviewer 1 Report

The authors have made substantial changes and adjustments to their article.

They have addressed each of the changes I requested step by step.

I recommend publication. Best regards.

Author Response

Dear Reviewer, 

Authors are very thankful for the comments, which help us to improve the manuscript. All changes proposed have been included in the manuscript and highlighted in yellow and track changes.  

Best Regards, 

Prof. Ramunas Antanaitis 

Question  

Answers  

The authors have made substantial changes and adjustments to their article.

They have addressed each of the changes I requested step by step.

I recommend publication. Best regards.

Thank you very much for your review and very good and helpful comments. 

Reviewer 3 Report

The new version of the manuscript entitled "Assessment of Ruminating, Eating, and Locomotion Behavior During Heat Stress Using Advanced Technological Monitoring has been significantly improved.

Conclusions should be based on the results obtained (line 403-405). Please correct.

Correct errors in citations references.

Author Response

Dear Reviewer, 

Authors are very thankful for the comments, which help us to improve the manuscript. All changes proposed have been included in the manuscript and highlighted in yellow and track changes.  

Best Regards, 

Prof. Ramunas Antanaitis 

Question  

Answers  

The new version of the manuscript entitled "Assessment of Ruminating, Eating, and Locomotion Behavior During Heat Stress Using Advanced Technological Monitoring” has been significantly improved.

Thank you very much for your suggestions and positive response

Conclusions should be based on the results obtained (line 403-405). Please correct.

We corrected conclusion section – we deleted – “We realized that innovative technologies such as RWS can be used to detect HS and its impact on reticulorumen parameters and cow walking activity”and added – “THI correlated negatively with RWS registered drinking time (r = 0.191, p< 0.05) and chews per bolus (r = 0.172, p < 0.01)”

Correct errors in citations references.

Corrected